# Enhancing Deep Imbalanced Regression via Frobenius Norm Regularization

## Abstract

Deep Imbalanced Regression (DIR) aims to train a deep neural network (DNN) model specified for the regression tasks from an imbalanced training distribution and generalize well on an unseen balanced testing distribution. While modern solutions have achieved significant progress in DIR, the performance of the samples still varies a lot across the different shots. For instance, the samples in the majority-shot always outperform the underrepresented (median and few-shot) samples, which motivates us to investigate whether we can leverage the well-trained majority-shot samples to help the other under-trained samples. Empirically, we observe that previous solutions in DIR often produce ordinal feature Frobenius norms across the majority-shot samples and considerably lower training Mean-Absolute-Error (MAE). Meanwhile, the underrepresented samples often violate the ordinality of the majority-shot Frobenius norms and exhibit a high training MAE. As a result, this demonstrates that compared to the majority-shot samples, the underrepresented samples are still under-fitted during the training process. More importantly, we can identify the training performance through the lens of the ordinality of the Frobenius norm. Motivated by this observation, we first analyze why the ordinality of the Frobenius norm can result in good training performance across the labels. Then, we introduce a feature regularization to encourage the feature Frobenius norms to be ordinal for all labels during the training process. Moreover, we propose a novel model training strategy that incorporates the knowledge from the well-trained majority samples to help the underrepresented samples. By training a linear model from the majority-shot samples to predict the feature Frobenius norm of underrepresented samples, we fine-tune the previously trained model to enhance the outcomes of underrepresented samples. Extensive experiments over the real-world datasets also validate the effectiveness of our proposed method. Anonymous code can be found in :Here

## 1 Introduction

Data Imbalance Regression (DIR) Yang et al. (2021), where certain labels are less observed or even missing during training the Deep Neural Networks (DNN) for the regression tasks, is one of the most intriguing problems in machine learning. Due to the imbalanced training distribution and the continuous label space, DIR poses new challenges in accurately regressing the target labels on the unseen balanced testing distributions. Recently, modern solutions have tried to solve the DIR from several perspectives. For instance, Yang et al. (2021) incorporated smoothing into both label and feature space within the loss functions. Gong et al. (2022) utilized ranking regularization to preserve the order of the feature space. Zha et al. (2023); Keramati et al. (2024) introduced contrastive learning to align the feature space with the label space. Pintea et al. (2023); Xiong & Yao (2024b); Pu et al. (2025) leveraged classification to help the DIR. Dong et al. (2025) utilized a geometric constraint to improve the quality of the representation feature for DIR.

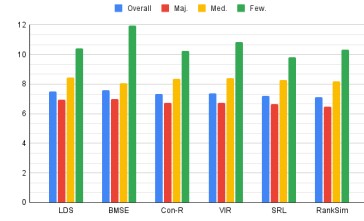

Figure 1: Per-method performance of each shot on AgeDB-DIR.

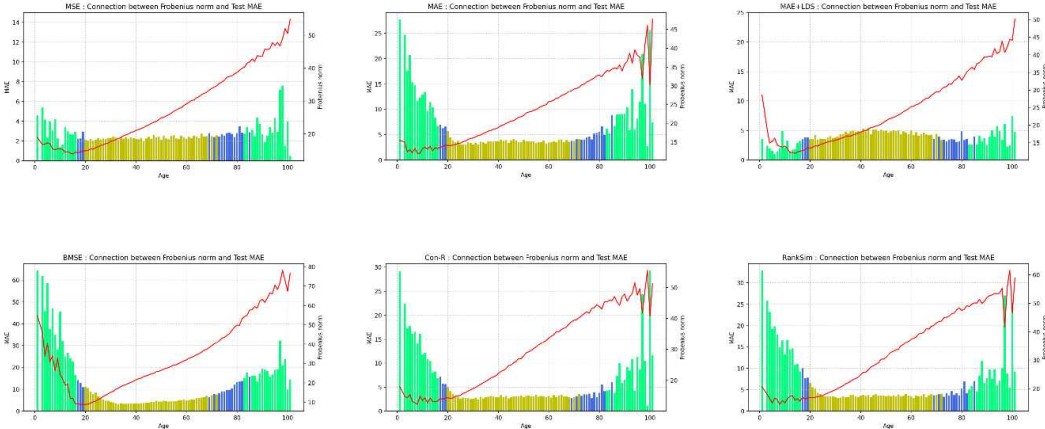

Figure 2: The illustration for the Frobenius norm of the feature and the MAE on AgeDB-DIR. From top left to right: **(1)** MSE, **(2)** MAE, and **(3)** MAE+LDS Yang et al. (2021). From bottom left to right: **(4)** BMSE Ren et al. (2022), **(5)** Con-R Keramati et al. (2024), and **(6)** RanskSim Gong et al. (2022). Bars (**Left axis**) denote the MAE for each training label, and the Red line (**Right axis**) denotes the Frobenius norm.

However, while previous works have achieved remarkable progress in tackling DIR, we can observe the performance divergence across the different shots of the data [1].

Specifically, as shown in Fig. 1, although significant performance improvements have been achieved by the previous methods, the performance of the few-shot and median-shot samples are significantly worse than that of the majority-shot samples. More specifically, we can observe that when we compare the overall performance with each shot, the majority-shot performance always contributes the most to the overall. In contrast, the others often overwhelm the overall performance. This phenomenon indicates that during the training process, the underrepresented samples (median/few shots) are also under-trained compared to the majority-shot samples. Therefore, it is necessary to explore the per-label performance in DIR. Considering the distribution shift from the imbalanced training distribution to the unseen balanced testing distribution in DIR, inspired by previous works Chen et al. (2021) where the scale of the Frobenius norm would significantly affect the performance of regression under the distribution shifts, we directly investigate the feature Frobenius norm for each training label with their testing Mean Absolute Error (MAE) during the training process.

Compared to the Domain Adaptation Regression (DAR) Chen et al. (2021); Nejjar et al. (2023); Adachi et al. (2025), DIR can be treated as one unique case of DAR where the testing distribution is balanced and the testing samples are not available during training. Therefore, to investigate the feature Frobenius norm (e.g., scales) for each label without access to the testing data (which made the subspace alignment unachievable Chen et al. (2021)), we observe the per-label Frobenius norm for the training samples. For easier interpretation, we calculate the Frobenius norm of each training label from its prototypes. As we can observe from Fig. 2, the previous methods can lead to an ordinality (linear in Fig. 2 (1,2,3,5,6) in Fig. 2 (4)) relationship between the scales of feature Frobenius norm across the majority-shot samples. However, they correspond to a considerably lower MAE across the training labels in Fig. 2 and lower testing MAE in Fig. 1, showcasing that these training labels in the majority-shot are well-fitted during the training process.

Interestingly, another phenomenon emerges in Fig. 2 that the feature Frobenius norm of the under-represented median and few-shot samples does not comply with the linear relationship as that of the majorities. Meanwhile, they often correspond with higher training MAE across the training labels, and their overall MAE is also higher than that of the majority-shot samples, which demonstrates that the training labels are still under-fitted compared to the majority-shot samples during training. On the other hand, our empirical observation also suggests that the ordinality in the Frobenius norm across the labels also shows strong correspondence to the training performance (e.g., higher train-

---

[1]Following Yang et al. (2021), we define the data with <20 samples as the few-shot samples (Few), >100 samples as the majority-shot samples (Majority), and others as the median-shot samples (Median).

ing MAE in Fig. 2 which violates this ordinality at underrepresented samples). Therefore, this motivates us to treat the ordinality of the feature Frobenius norm as an indicator to identify the training performance. Moreover, this also encourages us to investigate this ordinality across the labels and leverage the feature Frobenius norm from the majority-shot samples to help the learning of the underrepresented median and few-shot samples in DIR.

In this paper, we first conduct a simple analysis on why the sequential Frobenius norm of the feature representation would lead to good per-label training performance. We design a three-phase training to incrementally enhance the per-label training performance to produce an improved overall outcome. Therefore, motivated by the analysis and empirical observation, we propose a label-distance-based regularization to encourage the Frobenius norm of the feature representation to be ordinal at the first phase. Then, we introduce a linear neural network to learn the ordinality of the feature Frobenius norm from the majority-shot samples, and we utilize the linear network to predict the Frobenius norm of the median and few-shot samples in the second phase. Moreover, we fine-tune the previously trained models by leveraging the predicted Frobenius norm to further enhance the outputs for the underrepresented data samples in the last phase. Extensive experiments on the real-world datasets validate the effectiveness of our proposed method.

In summary, we conclude our contributions as follows:

- We empirically observe the correlation between the Frobenius norm and the training MAE in DIR, demonstrating that learning towards a sequential Frobenius norm across the labels can result in better training performance.

- We propose a label-distance-based regularization to encourage the model to maintain the ordinality of feature Frobenius norm across the labels, and we leverage the Frobenius norm of the majority-shot samples to train a Frobenius linear network to guide the median and few-shot samples.

- We incorporate the prediction from the Frobenius linear network to fine-tune the trained models to further address the DIR.

- Extensive experiments on three real-world datasets validate the effectiveness of our method.

## 2 RELATED WORKS

### 2.1 DEEP IMBALANCED REGRESSION

DIR was first studied in Yang et al. (2021) and has been widely discussed by recent research. SMOTER Torgo et al. (2013) was first proposed to use synthetic data to address the imbalance. Branco et al. (2018); Belhaouari et al. (2024) leveraged the oversampling strategy for the imbalanced regression tasks. However, they mostly handle the low-dimensional imbalance cases. Steininger et al. (2021); Yang et al. (2021) incorporated kernel-density estimation (KDE) to assign the smoothed weights to different labels. Liu et al. (2023) extended the work from Yang et al. (2021) in estimating the label density to deal with the fetal brain age regression task under the label. BMSE Ren et al. (2022) reviewed the DIR from a statistical view and proposed a balanced-MSE loss function to accommodate the imbalanced training label distribution, but its performance replies heavily on a balanced testing distribution.

Moreover, modern research has also investigated leveraging representation learning to address the DIR. Ranksim Gong et al. (2022) was the first work proposed to tackle the DIR by maintaining the label space relationship with the feature space. Meanwhile, Liu et al. (2023) introduced a ranking regularization for the feature representation to handle the label imbalance. Zha et al. (2023) incorporated contrastive learning to pull the samples against each other with different scales based on their relative distance. Lee (2023) tried to learn a feature space with a more balanced structure. Con-R Keramati et al. (2024) modeled global and local label similarities in the feature space and pushed the negative samples from the anchors with more penalties. Dong et al. (2025) also tried to use geometric constraints on the feature representation space (e.g., unit hypersphere) to tackle the distortion caused by the target label imbalance. Lim et al. (2025) leveraged learnable proxies to construct a balanced feature space for DIR. Although they have achieved impressive results in DIR, they often encounter heavy computational burdens. Furthermore, Zhang et al. (2023; 2024; 2025) proposed the

entropy-based method to regularize the feature representation. However, they mostly focused on the balanced regression tasks.

Specifically, it has been explored by the recent research Pu et al. (2025); Xiong & Yao (2024a); Pintea et al. (2023) that treating regression as classification tasks can effectively help to address the imbalance. Moreover, Lin et al. (2024) adopted the classification as an auxiliary task to provide additional gradients for median and few-shot samples through the multi-grained classifiers. However, the inherent difference between the classification and regression would still hinder the effectiveness of classification from solving the DIR (e.g., unbounded label space).

## 3 METHODOLOGY

In this paper, we try to take advantage of the majority shot to enhance the learning of median and few-shot samples by leveraging the feature Frobenius norm in three sequential training phase. Firstly, we propose an ordinal regularization to encourage the Frobenius norm of the feature representations to exhibit ordinal. Then, we utilize a linear neural network to learn the underlying correlation between the Frobenius norm and its corresponding target label with only majority-shot samples. Moreover, we employ this network to predict the Frobenius norm from the median and few-shot samples. By taking the predicted Frobenius norm of the median and few-shot samples to fine-tune the preciously trained models, we can leverage the information from the majority to help the median and few-shot samples to address the DIR.

### 3.1 PRELIMINARY AND NOTATIONS

In this paper, we have a $N_{tr}$-samples training set $D_{tr} = \{x_i, y_i\}_{i=1}^N$ where $x_i \in \mathcal{X}$ is the input data and $\mathcal{X}$ is the input space, and $y_i \in \mathcal{Y} = \mathbb{R}$ denotes the target label and $\mathcal{Y}$ is the label space [2]. Following Yang et al. (2021), we divide the continuous label space $\mathcal{Y}$ into $B$ bins with equal intervals, and each bin index represents a target label. Therefore, the maximum target label would be $y^B$. In DIR, the training set $D_{tr}$ follows an underlying highly skewed distribution $D_{tr} \sim P_{tr}$ while the testing set $D_{te}$ is balanced. We denote the deep neural network as $\{f, g\}$, where $f$ is the feature extractor and $g$ is the regressor. We first extract the feature representation $z_i = f(x_i)$ where $z_i \in \mathcal{R}^d$ with $d$ dimensions, and then we forward the feature to the regressor to obtain the target prediction $\hat{y}_i = g(z_i)$. Following Yang et al. (2021), we employ Mean Squared Error (MSE) as the loss function in the baseline.

### 3.2 MOTIVATION: A SIMPLE ANALYSIS OF THE REGRESSION TASK

For a feature representation $z$ and a scalar target label. For $\lambda \geq 0$, considering the ridge objective for the regression task $w \in \arg\min_{w \in \mathcal{R}^d}(z^\top w - y)^2 + \lambda\|w\|_2^2$, we can obtain the closed form of the minimized $w$ as $w = \frac{y}{\|z\|_2^2 + \lambda}z$. Since for a vector $z$, the Frobenius norm is equivalent to the Euclidean norm, we have $w = \frac{y}{\|z\|_F^2 + \lambda}z$. Therefore, for a regressor $g$ which aims to regress the features from all labels with its corresponding $w$, the scale of the feature Frobenius norm would significantly impact the $w$ with the growth of $y$. To ensure the generalization ability of $w$, which can be applied to all labels for a regressor, the scale of the $\|x\|_F^2$ should grow with the increase of the $y$ given the monotonic increase after $\|x\|_F^2 \geq \lambda$ in $\|w\|_2^2$. Meanwhile, this also coincides with the empirical observation in Fig. 2 and the enhanced outcomes from Fig. 1, where the ordinal Frobenius norm of the features in each label would lead to considerably low training errors, which encourages us to regularize on the Frobenius norm across the labels to tackle the DIR.

### 3.3 ORDINAL REGULARIZATION ON FEATURE FROBENIUS NORM: PHASE ONE

We propose a feature regularization to force the predicted features to be ordinal given their target labels. For a given training mini-batch that contains $m$ target labels, we first calculate the feature prototype of each label, e.g., $c^j = \frac{1}{b_j}\sum_{i=1}^{b_j} z_i^j$ where $z_i^j$ is the feature representation of the label

---

[2]Following Ren et al. (2022); Zha et al. (2023), we discuss the target regression label with 1 dimension in this paper for easier interpretation, but can still be generalized to multiple-labels scenarios.

$j$ and $b_j$ denotes the number of samples in label $j$ of the mini-batch $b$. Therefore, to achieve the agreement between the same labels and the feature prototypes, we first introduce an alignment loss to regularize across the same labels.

$$\mathcal{L}_{align} = \sum_{j \in [m]} \| z^j_\cdot - c^j \|_2^2 \tag{1}$$

where $z^j_\cdot$ denotes the samples belonging to the label $j$, and $[m]$ denotes the set of target labels in the mini-batch. Then, we propose a feature Frobenius norm regularization by penalizing the Frobenius norm with different scales based on their relative distance in the target label space.

Then, we can generate a prototype Frobenius norm and label pair $(y^j, l^j)_j^m$, and we regularize the $(l_j)_j^m$ based on the relative distance across the $(y^j)_j^m$. We try to align the Frobenius norm of the feature prototypes with the relative distance of their corresponding labels to maintain their ordinality. Since the direct regularization of the label distance in the feature space would result in discriminative representations Zeng et al. (2021), which would empirically lead to a discontinuous feature space under the DIR Pu (2025) and result in unsatisfying outcomes. Inspired by Hinton & Roweis (2002); Lim et al. (2025), we introduce an ordinal feature regularization from a distribution-matching perspective. Specifically, we force the feature Frobenius norms to approach the ordinal relationship as that of the label space based on the distribution over their relative distance. Therefore, we map the pairwise label distance into a distance probability distribution $p_{ij}$:

$$p_{ij} = \frac{e^{-d(y^i, y^j)}}{\sum e^{-d(y^\cdot, y^\cdot)}} \tag{2}$$

where $d(y^\cdot, y^\cdot)$ denotes the distance between arbitrary two target labels. Similarly, we can construct the feature prototype Frobenius norm probability distribution in the same manner: $q_{ij} = \frac{e^{-d(l^i, l^j)}}{\sum e^{-d(l^\cdot, l^\cdot)}}$. And we can regularize them with a Kullback-Leibler divergence:

$$\mathcal{L}_{KL} = \sum p_{ij} \log \frac{p_{ij}}{q_{ij}} \tag{3}$$

However, as stated in Zhang et al. (2023), the alignment loss in Eq. 1 is equivalent to minimizing the MSE, e.g., $\mathcal{L}_{MSE} \propto \mathcal{L}_{align}$. Therefore, we implement the MSE instead to reduce the computational burden. Moreover, to avoid feature collapse in Eq. 3, we incorporate the ordinal entropy (OE) Zhang et al. (2023) to encourage the feature prototypes of different labels to spread far away from each other in the feature space.

$$\mathcal{L}_{OE} = -\frac{1}{m(m-1)} \sum_i \sum_{j \neq i} d(i,j) \| c^i - c^j \|_2^2 \tag{4}$$

where $d(i, j)$ denotes the L1 distance between the label $y^i$ and $y^j$.

In summary, the final ordinal regularization can be formulated as :

$$\mathcal{L}_{ordinal} = \mathcal{L}_{KL} + \alpha \mathcal{L}_{OE} \tag{5}$$

where $\alpha$ is a hyper-parameter that serves as the trade-off between the two losses. Therefore, we can incorporate the above loss $\mathcal{L}_{ordinal}$ in the training phase with a hyper-parameter $\beta$ as:

$$\mathcal{L}_{training} = \mathcal{L}_{any} + \beta \mathcal{L}_{ordinal} \tag{6}$$

where $\mathcal{L}_{any}$ denotes any loss penalty in addressing the DIR, e.g., LDS Yang et al. (2021).

### 3.4 MAJORITY-GUIDED LINEAR PROTOTYPE NETWORKS: PHASE TWO

After we have obtained the trained model from the above loss Eq. 6, we can then exploit the well-fitted majority to help the under-fitted minority. As we can observe from Fig. 3, it is obvious that the majority-shot samples perform a lot better than the other shots. Meanwhile, the majority shot performance reaches convergence at a very early phase, e.g., epoch $\approx 40$ while the rest of the training mostly focuses on optimizing the median and few-shot samples, but their efforts exhibit trivial. Therefore, this motivates us to take advantage of the well-fitted majority-shot samples.

To leverage information from the majority-shot samples of trained models and aid the median and few-shot samples in DIR, motivated by our empirical observations, we employ the Frobenius norm of the majority samples to guide the learning of the median and few-shot samples. To calculate the per-label Frobenius norm across the different training labels, we employ the feature prototype of each label as the anchor to help the underrepresented samples.

Once we have obtained the prototypes of each training label $C = (c^1, c^2, \ldots, c^B)$ where $B$ is the maximum training label in the training set, we then compute their corresponding Frobenius norm (e.g., $l^j = \|c^j\|_F = \sqrt{\sum_i^d e_i^j}$ where $e_i^j$ denotes the $i$-th element in $c^j$) and construct a set of $n$ label-norm pairs $s_{maj} = (y^j, l^j)$ where each $j \in y^{maj}$ is the majority-shot labels $y^{maj}$.

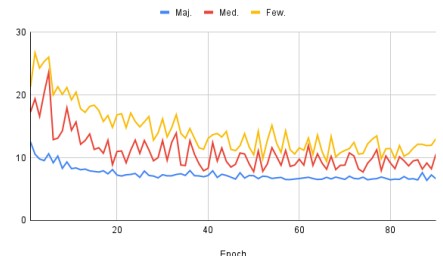

One advantage of only employing majority-shot samples to construct the label-norm pairs is that we do not encounter the label-missing problem at this step. Meanwhile, since the majority-shot samples are well-trained compared to the underrepresented median and few-shot samples (Fig. 3), they could provide more reliable supervision signals ($l \in s_{maj}$) than the underrepresented samples.

Figure 3: Training MAE for per-label for different shots on AgeDB-DIR with MSE.

Therefore, to learn from the majority-shot samples, we feed the $s_{maj}$ into a Frobenius neural network $f_F$ to learn the tendency of the Frobenius norm across the majority-shot samples. Considering that the individual label only corresponds to one unique pair in the label-norm set (e.g., for a dataset with $B$ labels, we only utilize $B$ pairs for training the $f_F$), we set $f_F$ as a one-layer linear neural network to avoid overfitting. Moreover, to guarantee the non-negativity of the predicted Frobenius norm during the training, we incorporate the Sigmoid function $\sigma$ into the output layer of the linear neural network $f_F$. Therefore, we can obtain the prediction of the Frobenius norm $\hat{l}^j = \sigma(f_l(y^j))$ for the given target label $y_j$, and we adopt the MSE: $\mathcal{L}_{f_F} = \|\hat{l}^j - l^j\|_2^2$ as the loss function for training this Frobenius norm prediction network $f_F$ from its ground truth $l^j$. Note that only the labels in the majority-shot samples are used to train the $f_F$, and the parameters of the trained models are kept frozen while the gradient of $\mathcal{L}_{f_F}$ only updates $f_F$ at this phase.

### 3.5 FINE-TUNING THE PRETRAINED MODELS: PHASE THREE

After we have obtained the Frobenius linear neural network $f_F$, we feed all labels into $f_F$ to get their predicted Frobenius norm $\hat{l}$. Therefore, we have a set of feature Frobenius norm predictions set $s = (y^j : \hat{l}^j)_{j=1}^B$ pairs. In the fine-tuning phase, when one data sample $x_i$ with label $j$ is fed into the feature encoder $f$, we can obtain its feature $z_i^j = f(x_i)$. Then, we calculate the Frobenius norm of the feature as $\|z_i^j\|_F$. We regularize the $\|z_i^j\|_F$ from the predicted Frobenius set $s$ as follows:

$$\mathcal{L}_{align}(z_i^j) = \|\hat{l}^j - \|z_i^j\|_F\|_2^2 \qquad (7)$$

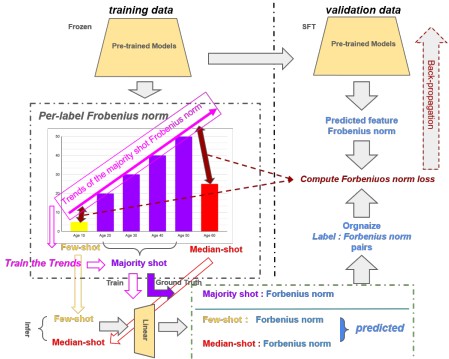

Figure 4: The post-hoc training step (Phase 2&3) of our proposed method.

and we regress the $z_i^j$ on the regressor $g$ and adopt MSE to penalize the regression outcomes as in Fig. 5. During the fine-tuning phase, only the parameters of the trained model $(f, g)$ are updated with the $\mathcal{L}_{align}$ while the parameters of the linear network $f_L$ are kept frozen. In summary, we first train with the $\mathcal{L}_{training}$ for the $(f, g)$ in more epochs $e_{main}$, and then we utilize the $\mathcal{L}_{align}$ to train the $f_F$ with fewer epochs $e_F$. Finally, we retrain the $(f, g)$ with $e_{SFT}$ epochs. The total training duration is $e_{main} + e_F + e_{SFT}$. We illustrate the assignment of each epoch in the next section.

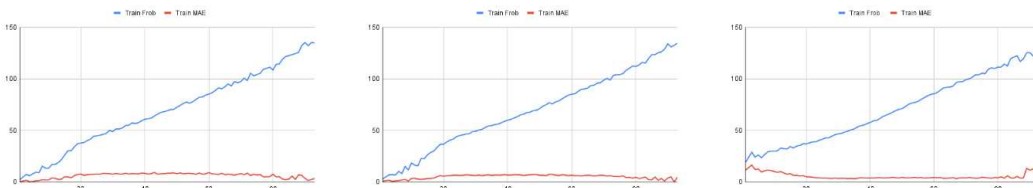

Figure 5: Demonstration of the improved Frobenius norm of the feature and the MAE on AgeDB-DIR. From top left to right: **(1)** BMSE Ren et al. (2022), **(2)** RankSim Gong et al. (2022) and **(3)** Con-R Keramati et al. (2024). Blue line : Frobenius norm. Red line : MAE per-label. We can observe the head and tail Frobenius norm comply with the ordinality of the majority-shot samples.

## 4 EXPERIMENTS

### 4.1 IMPLEMENTATION DETAILS

We implement our experiments on three real-world datasets, AgeDB-DIR, IMDB-WIKI-DIR, and STS-B-DIR. AgeDB-DIR and IMDB-WIKI-DIR are the tasks of age regression from facial images. STS-B-DIR is the task of similarity regression from pairs of sentences. We adopt the ResNet-18 as the backbone for the AgeDB-DIR, and ResNet-50 as the backbone for the IMDB-WIKI-DIR. Also, we used the BiLSTM + GloVe word embeddings and preprocessed them for STS-B-DIR. In this paper, we adopt the MAE as the loss penalty for the regression prediction, which is also consistent with the Yang et al. (2021).

**Baselines** We adopted the following baselines to make the comparison with our proposed method: SMOTER Torgo et al. (2013), SMOGN Branco et al. (2017), RRT Kang et al. (2020), FOCAL-R Lin et al. (2017), LDS and FDS Yang et al. (2021), DER Amini et al. (2020), VAE Kingma & Welling (2013), RANKSIM Gong et al. (2022) , OE Zhang et al. (2023), OE Zhang et al. (2023), Con-R Keramati et al. (2024), SupCR Zha et al. (2023), and SRL Dong et al. (2025).

### 4.2 DATASETS

We validate the effectiveness of our proposed method on three real-world datasets following Yang et al. (2021), which include both visual tasks and natural language processing tasks, to provide the performance evaluation:

**AgeDB-DIR** is one real-world dataset that aims to regress the age from facial images. It was constructed by Moschoglou et al. (2017) and then re-organized by Yang et al. (2021) to accommodate the DIR. It contains 12.2K images of training data, 2.1K images of validation data, and 2.1K images of testing data. The bin length is set to 1 year (the minimum resolution), the minimum age is 0, and the maximum age is set to be 101 (following Yang et al. (2021), and the centroids of the bins are set to be the target labels).

**IMDB-WIKI-DIR** is another real-world dataset with the goal of regressing ages from the facial images. It was constructed by Rothe et al. (2018) and re-organized by Yang et al. (2021) for DIR. It contains 235K face images in total. Following Yang et al. (2021), 191.5K facial images are used for training, 11K images are used for validation, and 11K images are used for the testing set. Among them, the validation set and testing set are considerably more balanced than the training set. The minimum resolution is also set to be 1 year for the AgeDB-DIR.

**STS-B-DIR** is a text similarity score dataset; the task is to regress the similarity of two paired sentences. It was constructed by Wang et al. (2018) and re-ordered by Yang et al. (2021). This dataset is collected from multiple sources, e.g., news headlines, videos, image captions, and natural language inference data etc. The dataset consists of a set of sentence pairs (embedding), which are annotated with an average similarity score, where the range of scores is from 0 to 5, and the minimum resolution is 0.1 for each target label. There are 5.2K pairs of training samples, 1K pairs of validation samples, and 1K pairs of testing samples. Similar to the previous datasets, the training distribution is highly imbalanced, while the validation and testing distributions are more balanced.

Table 1: Evaluation on AgeDB-DIR.

| Shot | MAE↓ | | | | GM↓ | | | |
|---|---|---|---|---|---|---|---|---|
| Method | All | Many. | Med. | Few. | All | Many. | Med. | Few. |
| VANILLA Yang et al. (2021) | 7.77 | 6.62 | 9.55 | 13.67 | 5.05 | 4.23 | 7.01 | 10.75 |
| SMOTER Yang et al. (2021) | 8.16 | 7.39 | 8.65 | 12.28 | 5.21 | 4.65 | 5.69 | 8.49 |
| SMOGN Yang et al. (2021) | 8.26 | 7.64 | 9.01 | 12.09 | 5.36 | 4.90 | 6.19 | 8.44 |
| RRT Yang et al. (2021) | 7.74 | 6.98 | 8.79 | 11.99 | 5.00 | 4.50 | 5.88 | 8.63 |
| FOCAL-R Yang et al. (2021) | 7.64 | 6.68 | 9.22 | 13.00 | 4.90 | 4.26 | 6.39 | 9.52 |
| SQINV Yang et al. (2021) | 7.81 | 7.16 | 8.80 | 11.20 | 4.99 | 4.57 | 5.73 | 7.77 |
| SQINV + LDS Yang et al. (2021) | 7.67 | 6.98 | 8.86 | 10.89 | 4.85 | 4.39 | 5.80 | 7.45 |
| LDS+FDS Yang et al. (2021) | 7.55 | 7.01 | 8.24 | 10.79 | 4.72 | 4.36 | 5.45 | 6.79 |
| LDS+FDS+DER Wang & Wang (2023) | 8.18 | 7.44 | 9.52 | 11.45 | 5.30 | 4.75 | 6.74 | 7.68 |
| VAE Kingma & Welling (2013) | 7.63 | 6.58 | 9.21 | 13.45 | 4.86 | 4.11 | 6.61 | 10.24 |
| RANKSIM Gong et al. (2022) | 7.02 | 6.49 | 7.84 | 9.68 | 4.53 | 4.13 | 5.37 | 6.89 |
| OE Zhang et al. (2023) | 7.46 | 6.73 | 8.18 | 12.38 | 4.72 | 4.21 | 5.36 | 9.70 |
| Con-R Keramati et al. (2024) | 7.20 | 6.50 | 8.04 | 9.73 | 4.59 | 3.94 | 4.83 | 6.39 |
| VIR Wang & Wang (2023) | 6.99 | 6.39 | 7.47 | 9.51 | 4.41 | 4.07 | 5.05 | 6.23 |
| SupCR Zha et al. (2023) | 6.85 | 6.20 | 7.62 | 10.82 | 4.32 | 3.89 | 4.95 | 8.02 |
| Distloss Nie et al. (2025) | 7.55 | 7.28 | 7.66 | 9.80 | 4.70 | 4.52 | 4.80 | 6.29 |
| SRL Dong et al. (2025) | 7.22 | 6.64 | 8.28 | 9.81 | 4.50 | 4.12 | 5.37 | 6.29 |
| Ours | 7.34 | 7.35 | **6.63** | **9.24** | 4.71 | 4.83 | **3.98** | **5.84** |

## 4.3 ANALYSIS ON THE RESULTS OF AGEDB-DIR

As we can observe from Tab. 1, our method establishes new state-of-the-art (SOTA) performance on median and few-shot samples, achieving MAE of 6.63 and GM of 3.98, both representing the best results across all compared methods. This performance represents a 13.0% improvement in MAE compared to the next best method (VIR Wang & Wang (2023): 7.47) and demonstrates the effectiveness of our approach in the intermediate regime between data-rich and data-scarce scenarios.

In the meantime, our method achieves remarkable performance on few-shot samples, attaining an MAE of 9.24, which represents a substantial improvement over competing methods. Compared to the second-best performer VIR Wang & Wang (2023) (9.51), our approach reduces the error by 2.8%. More significantly, when compared to the overall best-performing method SupCR Zha et al. (2023), our approach achieves a 14.6% relative improvement on few-shot samples (9.24 vs 10.82), demonstrating superior capability in handling the most challenging cases where training data is severely limited.

On many-shot samples, our method achieves MAE of 7.35, which is higher than top-performing methods such as SupCR Zha et al. (2023) (6.20) and VIR Wang & Wang (2023) (6.39). This result, on the other hand, represents a strategic trade-off where our method sacrifices some performance on data-abundant samples to achieve superior performance on challenging cases. This trade-off is theoretically motivated and practically valuable, as real-world applications often require robust performance across all frequency regimes rather than optimization for the majority class alone.

Moreover, we also integrate our method with MSE, BMSE Ren et al. (2022), Con-R Keramati et al. (2024), and RankSim Gong et al. (2022), in Fig. 5. We can observe that our method can lead to a smoother feature Frobenius norm in the regions of median and few-shot samples. Meanwhile, the overall Frobenius norm follows strictly the ordinality in the feature Frobenius norm across different methods, and they also exhibit lower training MAE compared to the Fig. 2, showcasing a better performance in fitting the training labels.

## 4.4 ANALYSIS ON THE RESULTS OF IMDB-WIKI-DIR

Our proposed method in Tab. 2 demonstrates strong performance in addressing DIR, achieving an overall MAE of 7.66, which positions it competitively among SOTA approaches while excelling in critical performance dimensions. Notably, our approach achieves the best performance on medium-shot samples (MAE: 9.67, GM: 6.36), significantly outperforming VIR's previous best of 11.81 MAE , demonstrating consistent superiority in data-scarce and moderately-represented scenarios. In addition, our method establishes new SOTA results on few-shot samples with an MAE of 19.25,

Table 2: Evaluation on IMDB-WIKI-DIR.

| Shot | MAE↓ | | | | GM↓ | | | |
|---|---|---|---|---|---|---|---|---|
| Method | All | Many. | Med. | Few. | All | Many. | Med. | Few. |
| BMC Ren et al. (2022) | 8.08 | 7.52 | 12.47 | 23.29 | - | - | - | - |
| GAI Ren et al. (2022) | 8.12 | 7.58 | 12.27 | 23.05 | - | - | - | - |
| VAE Kingma & Welling (2013) | 8.04 | 7.20 | 15.05 | 26.30 | 4.57 | 4.22 | 10.56 | 20.72 |
| RANKSIM Gong et al. (2022) | 7.50 | 6.93 | 12.09 | 21.68 | 4.19 | 3.97 | 6.65 | 13.28 |
| DER Amini et al. (2020) | 7.85 | 7.18 | 13.35 | 24.12 | 4.47 | 4.18 | 8.18 | 15.18 |
| LDS + FDS + DER Wang & Wang (2023) | 7.24 | 6.64 | 11.87 | 23.44 | 3.93 | 3.69 | 6.64 | 16.00 |
| Con-R Keramati et al. (2024) | 7.33 | 6.75 | 11.99 | 22.22 | 4.02 | 3.79 | 6.98 | 12.95 |
| VIR Wang & Wang (2023) | 7.19 | 6.56 | 11.81 | 20.96 | 3.85 | 3.63 | 6.51 | 12.23 |
| Distloss Nie et al. (2025) | 7.80 | 7.21 | 12.60 | 23.33 | 4.45 | 4.18 | 7.71 | 15.43 |
| SRL Dong et al. (2025) | 7.69 | 7.08 | 12.65 | 22.78 | 4.28 | 4.03 | 7.28 | 15.25 |
| Ours | 7.66 | 7.37 | **9.67** | **19.25** | 4.31 | 4.11 | **6.36** | **12.09** |

Table 3: Evaluation on STS-B-DIR.

| Shot | MSE↓ | | | | Pearson Correlation↑ | | | |
|---|---|---|---|---|---|---|---|---|
| Method | All | Many. | Med. | Few. | All | Many. | Med. | Few. |
| LDS Yang et al. (2021) | 0.914 | 0.819 | 1.319 | 0.955 | 75.6 | 73.4 | 63.8 | 76.0 |
| FDS Yang et al. (2021) | 0.927 | 0.851 | 1.225 | 1.012 | 75.0 | 72.4 | 66.7 | 74.2 |
| INV Yang et al. (2021) | 1.298 | 1.300 | 1.281 | 1.319 | 62.8 | 60.3 | 59.6 | 66.3 |
| VAE Kingma & Welling (2013) | 0.968 | 0.833 | 1.511 | 1.102 | 75.1 | 72.4 | 62.1 | 74.0 |
| VIR Wang & Wang (2023) | 0.892 | 0.795 | 0.899 | 0.781 | 77.6 | 75.2 | 69.6 | 84.5 |
| LDS + FDS Yang et al. (2021) | 0.907 | 0.802 | 1.363 | 0.942 | 76.0 | 74.0 | 65.2 | 76.6 |
| RANKSIM Gong et al. (2022) | 0.903 | 0.908 | 0.911 | 0.804 | 75.8 | 70.6 | 69.0 | 82.7 |
| DER Amini et al. (2020) | 1.001 | 0.912 | 1.368 | 1.055 | 73.2 | 71.1 | 64.6 | 74.0 |
| LDS + FDS + DER Wang & Wang (2023) | 1.007 | 0.880 | 1.535 | 1.086 | 72.9 | 71.4 | 63.5 | 73.1 |
| SRLDong et al. (2025) | 0.877 | 0.886 | 0.873 | 0.745 | 76.5 | 70.8 | 74.9 | 84.4 |
| Ours | 0.895 | 0.899 | **0.869** | 0.912 | 76.7 | 72.3 | **75.6** | 81.3 |

representing a 4% improvement over the previous best method VIR Wang & Wang (2023) and more substantial gains over other competing approaches such as RANKSIM Gong et al. (2022) and Con-R Keramati et al. (2024). The consistent pattern of superior few-shot and medium-shot performance across both AgeDB-DIR and IMDB-WIKI-DIR datasets validates the generalizability and robustness of our approach for imbalanced regression tasks, particularly in scenarios where minority class accuracy is paramount for real-world deployment.

### 4.5 ANALYSIS ON THE RESULTS OF STS-B-DIR

Our proposed method in Tab. 3 achieves competitive performance with an overall MSE of 0.895 and Pearson correlation of 76.7%, demonstrating effective handling of imbalanced regression in semantic similarity tasks, representing a significant 8.2% improvement in correlation for this challenging intermediate regime on medium-shot samples. It illustrates consistent strength across different sample frequency regimes without the severe performance degradation observed in traditional approaches like LDS and FDS on medium-shot samples. This consistent pattern of medium-shot excellence across three diverse datasets (AgeDB-DIR, IMDB-WIKI-DIR, and STS-B-DIR) demonstrates the generalizability of our approach for imbalanced regression tasks spanning computer vision and natural language processing domains.

**More experiment details and the ablation study can be found in the supplementary materials.**

### 5 CONCLUSION

In this paper, we investigate the Frobenius norm in enhancing the performance of the models in DIR. We propose a three-phase training where each phase corresponds to a distinct loss or additional model architecture, which aims to exploit the feature Frobenius norm from the well-trained Frobenius norm to help the underrepresented samples. Extensive experiments over the real-world datasets also validate the effectiveness of our proposed method.

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

## A   USE OF LARGE LANGUAGE MODELS (LLMS)

We declare that Large Language Models (LLMs) were used exclusively as auxiliary tools to aid the writing of this paper, primarily for polishing, grammar checking, and improving readability. LLMs were not involved in conceptual design, theoretical formulation, experimental implementation, or result analysis. All research contributions, including methodology, experiments, and conclusions, were conceived, conducted, and validated entirely by the authors.

