# ENHANCING DEEP IMBALANCED REGRESSION VIA FROBENIUS NORM REGULARIZATION

## ABSTRACT

Deep Imbalanced Regression (DIR) aims to train a deep neural network (DNN) model specified for the regression tasks from an imbalanced training distribution and generalize well on an unseen balanced testing distribution. While modern solutions have achieved significant progress in DIR, the performance of the samples still varies a lot across the different shots. For instance, the samples in the majority-shot always outperform the underrepresented (median and few-shot) samples, which motivates us to investigate whether we can leverage the well-trained majority-shot samples to help the other under-trained samples. Empirically, we observe that previous solutions in DIR often produce ordinal feature Frobenius norms across the majority-shot samples and considerably lower training Mean-Absolute-Error (MAE). Meanwhile, the underrepresented samples often violate the ordinality of the majority-shot Frobenius norms and exhibit a high training MAE. As a result, this demonstrates that compared to the majority-shot samples, the underrepresented samples are still under-fitted during training and the ordinality of the Frobenius norm can also be treated as an indicator to identify the training performance. Motivated by this observation, we first analyze why the ordinality of the Frobenius norm can result in good training performance across the labels. Then, we introduce a feature regularization to encourage the feature Frobenius norms to be ordinal for all labels during the training process. Moreover, we propose a novel model training strategy that incorporates the knowledge from the well-trained majority samples to help the underrepresented samples. By training a linear model from the majority-shot samples to predict the feature Frobenius norm of underrepresented samples, we fine-tune the previously trained model to enhance the outcomes of underrepresented samples. Extensive experiments over the real-world datasets also validate the effectiveness of our proposed method. Code can be found in :Here

# 1 APPENDIX

## 1.1 DISCUSSIONS

**Distance metrics.** In the loss function $\mathcal{L}_{KL}$, the distance function which aims to measure the distance between the two arbitrary labels is defined as $d(\cdot, \cdot)$. Since the elements in $d(\cdot, \cdot)$ is always two integer, it is not possible to use the cosine similarity in $d$. We adopt MAE in our main paper, the reason why we did not use MSE is that the MSE would enlarge the distance of $d$ in the quadratic scale. Therefore, it would lead the values in distance set with huge difference and consequently make the probability distribution sparse, which limits the effectiveness of utilizing distribution divergence.

**More implementation details.** For the first phase, we use the learning rate as 1e-3 with weight decay. For the second phase, we use the learning rate as 1e-4 and we use 5e-5 as the learning rate for the phase three. To make a fair comparison between our method and other works, following Yang et al. (2021), we use the MAE and Geometric Mean (GM) to evaluate the AgeDB-DIR and IMDB-WIKI-DIR. We use the MSE and Pearson Correlation to evaluate the STS-B-DIR. We report results for the four subsets: All, Many, Median, and Few. for each dataset to identify the effectiveness of our method.

**Discussion on our method and the Lim et al. (2025).** Compared to Lim et al. (2025), although we both used the probability distribution alignment to achieve the ordinality, but our work differs a lot. Firstly, our work differs from Lim et al. (2025) in the motivation, based on our empirical observation, we try to encourage the feature Frobenius norm to maintain the ordinal as the majority-shot samples while Lim et al. (2025) tried to regularize on the feature space. Secondly, Lim et al. (2025) introduced the additional parameters to the model (e.g. ResNet) training in implementation. Therefore, our method is distinct from Lim et al. (2025).

## 1.2 ABLATION STUDY ON DIFFERENT SETTING OF THE EPOCH

To investigate the impact of the epoch settings across the three phases, we provide an ablation study on the different settings of the epoch on the regression model training (e.g., the ResNet-18 on AgeDB-DIR) $e_{main}$, the Frobenius linear model $f_F$ training $e_F$ and the fine-tuning phase on the previously trained main model (e.g., the ResNet-18 on AgeDB-DIR) $e_{SFT}$. To make a fair comparison, we set the total number of training epochs $e_{total} = e_{main} + e_F + e_{SFT}$ as the $e = 100$, which is the same experimental setting as the previous methods. To conduct the ablation study on the three-phase epoch, we strictly constrain the total training epoch to be $e = 100$, e.g., $e_{main} = 70$, $e_F = 20$ and $e_{SFT} = 10$. To better observe the varying of MAE across different epochs, we provide the ablation study on the setting of different epochs in Fig. 1 for AgeDB-DIR and Fig. 2 for IMDB-WIKI-DIR.

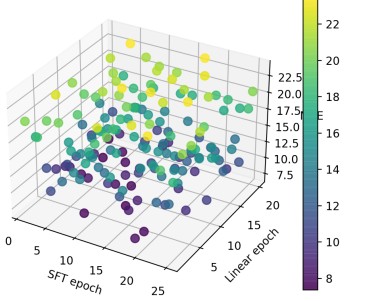 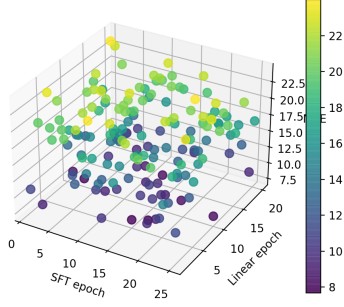

Figure 1: Ablation study on the different setting of epochs on AgeDB-DIR    Figure 2: Ablation study on the different setting of epochs on IMDB-WIKI-DIR

As we can observe from Fig. 1 and Fig. 2, higher $e_F$ and $e_{SFT}$ do not necessarily produce a better MAE. This is because that the the higher $e_F$ and $e_{SFT}$ are, the lower $e_{main}$ is. As a result, the main model is not well-trained. Meanwhile, the shallow linear Frobenius network model would easily

overfit with the high $e_F$. Therefore, the feature Frobenius norm they output would not have a high quality, which would then affect the performance of the latter phase. However, the lower $e_F$ and $e_{SFT}$ are also not guarantee a good overall MAE performance. The reason for this is that when the main model is well-trained at the Phase 1, there would leave less time for the $f_F$ and fine-tuning. Consequently, they would have a higher probability of getting under-trained, especially for the Phase 3 which fine-tunes the main model.

### 1.3   ABLATION STUDY ON HYPER-PARAMETERS : $\alpha$ AND $\beta$.

The hyper-parameter in our method is the trade-off between the $\mathcal{L}_{KL}$ and $\mathcal{L}_{OE}$ : $\alpha$. The goal of this hyper-parameter is to balance the impact of maintaining the ordinal feature Frobenius norm and the discriminative between different feature prototypes. Higher $\alpha$ would lead the model to focus more on learning discriminative feature prototypes, while lower $\alpha$ would lead the model to learn more ordinal information for the feature Frobenius norm. The hyper-parameter $\beta$ aims to balance the ordinal penalty with the regression loss (e.g., MAE in our main paper). Higher $\beta$ would lead the model to focus more on the ordinal characteristics while lower $\beta$ would make the model difficult to obtain the ordinality. We provide the ablation study on the hyper-parameters in Fig. 1.3.

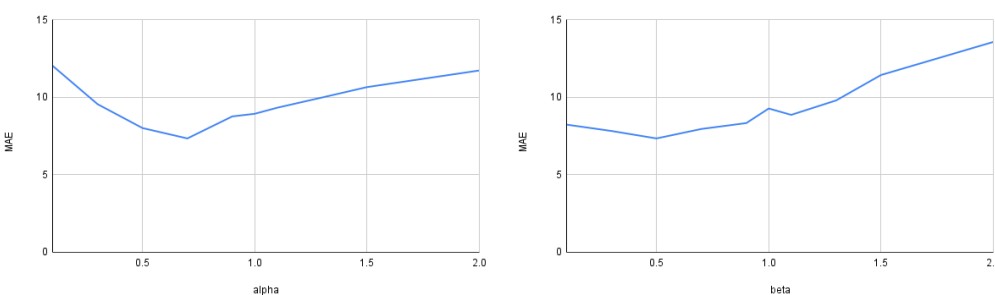

Figure 3: Ablation study on the different $\alpha$ on AgeDB-DIR

Figure 4: Ablation study on the different $\beta$ on AgeDB-DIR

### 1.4   ABLATION STUDY ON EACH PART OF THE TRAINING PHASE.

In this section, we compare the effectiveness between different training sections. Specifically, we compare the performance of four models on AgeDB-DIR as follows: vanilla model (MAE loss only), the ordinal model (MAE + ordinal without fine-tune), the vanilla model with fine-tune (MAE + with fine-tune on majority shots), and ordinal model with fine-tune (MAE + ordinal + with fine-tune on majority shots : Our method). We provide the ablation study on the effectiveness of each training phase in Fig. 1.4. We can observe that the although each part of the training phase can somewhat improve the performance, the enhancement is very limited. The reason why each separate training phase can not significantly improve the performance is: only the ordinality in the Frobenius norm does not have a direct impact on the regression result. In the meantime, merely using the

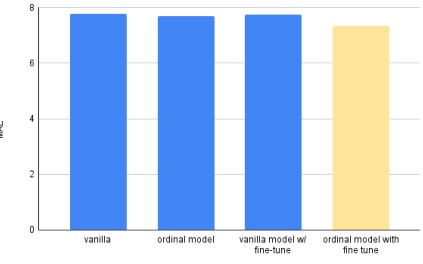

Figure 5: The effectiveness of each training phase on AgeDB-DIR.

majority-shot samples would easily make the trained model to over-fit on the majority-shot and cannot improve the performance on the underrepresented samples. Therefore, all of the training phases are necessary in our proposed method.