# OpenReview forum: "Enhancing Deep Imbalanced Regression via Frobenius Norm Regularization"
_ICLR.cc/2026/Conference — Submitted to ICLR 2026_

### Official Review · Reviewer_6WnH · 2025-10-29

**Soundness:** 3
**Presentation:** 3
**Contribution:** 2
**Rating:** 2
**Confidence:** 5

**Summary:**

This paper proposes a three-phase method for Deep Imbalanced Regression that (i) regularizes feature Frobenius norms to be ordinal across labels, (ii) learns a linear “F-norm predictor” from majority-shot prototypes, and (iii) fine-tunes the model to align minority examples’ feature norms to those predictions. Experiments on AgeDB-DIR, IMDB-WIKI-DIR, and STS-B-DIR report gains—especially on medium/few shots—while sometimes trading off many-shot accuracy. I currently see the contribution as interesting but incremental, with theoretical justification and evaluation rigor not yet at the level I expect for acceptance.

**Strengths:**

There is a lot of things I like about this paper and I need to commend Authors for:

+ The proposed pipeline is clear and modular (ordinal regularization → majority-guided linear predictor → student fine-tuning), should be easy to implement and there is a clear reason why it boosts performance.
+ The offered analysis connecting ridge regression scaling to a monotonic relation between label magnitude and feature norm is intuitive and motivates the norm-ordering prior.
+ I appreciate the per-shot reporting and the explicit emphasis on medium/few shots; the narrative acknowledges a conscious trade-off on many-shot accuracy.
+ Breadth across three datasets (two vision, one NLP) with concrete architecture details (ResNet-18/50; BiLSTM+GloVe) helps with generalization claims.

**Weaknesses:**

However, I see multiple weak points in this paper, that while not being disqualifying for the presented approach, inhibit the potential of this work to be published in the top venue such as ICLR:

+ The core ideas—prototype alignment, ordinal/entropy regularization, and rank/contrastive structure—feel close to prior lines (e.g., ranking, ordinal entropy). The “majority-guided linear predictor” is a small post-hoc head on top of prototypes. I’m missing a sharper conceptual advance beyond combining known ingredients.
+ The ridge-based argument does not establish that enforcing global ordinal F-norms improves test MAE under imbalance; it also ignores invariances (e.g., rescaling features vs. regressor). The leap from “larger ∥z∥ correlates with label magnitude” to “make ∥z∥ globally monotone” is under-theorized.
+ The method bins the continuous label space into B bins and computes mini-batch prototypes for losses. This introduces discretization noise and batch-dependence that may undermine the continuous-label objective the paper aims to improve.
+ The three-phase schedule adds extra stages and epochs, yet the paper does not quantify wall-clock or compute overhead vs. baselines, an important consideration for ICLR-caliber claims.
+ Reported improvements concentrate on medium/few shots, while many-shot degrades on AgeDB; the paper frames this as a “strategic trade-off,” but the real-world desirability depends on application. I am missing a thorough analysis of when that trade-off is net-positive.
+ No statistical variability (seeds/CI) is reported; conclusions rely on single numbers and qualitative norm plots.
+ For STS-B-DIR, the backbone is BiLSTM+GloVe rather than modern Transformers, weakening the generality claim on NLP.
+ The paper toggles between MSE (method description) and MAE (implementation) as the main prediction loss, causing confusion about the training objective.

**Questions:**

I would like for the Authors to address the following questions and I hope that the follow-up discussion will be useful for the Authors, even if helping to improve the paper for the next submission:

+ Can you provide a theoretical guarantee (or at least a generalization bound/assumption set) under which enforcing global ordinal Frobenius norms provably reduces test MAE in DIR, beyond the ridge-scaling heuristic?
+ How sensitive is the approach to the choice of binning (B) and to mini-batch prototype noise? Have you tried dataset-level prototypes or EMA prototypes to stabilize the losses?
+ Please report compute/time and multiple-seed results (mean±std) for all datasets, and discuss the practical trade-off w.r.t. methods that do not require post-hoc phases.
+ For AgeDB-DIR where many-shot degrades, can you quantify when the few/medium gains outweigh this loss (e.g., a cost-weighted metric or application-motivated utility)?
+ On STS-B-DIR, would the conclusions hold with a Transformer encoder (e.g., DeBERTa/BERT) under the same imbalance protocol?

---

> ### Author Response · Authors · 2025-12-03
> **Rebuttal for Reviewer  6WnH**
>
> **W1 : ...missing a sharper conceptual advance beyond combining known ingredients**
> The main contribution of this paper is to enhance the overall performance by improving the median and few-shot sample performance. From Fig. 1, we have observed that the connection between the Frobenius norm of the features on each label and the test MAE, where we observed that following the ordinal in the Frobenius norm on each label corresponds to the low MAE. Moreover, we also conducted a qualitative analysis of the relationship between the feature Frobenius norm and the last layer weights. Together, we aim to regularize the ordinal of feature Frobenius norm to enhance the outcomes of the models.
>
> Specifically, we designed a three-stage training phase. To begin with, the first phase is to force all features to maintain ordinal on their Frobenius norm. Compared to ordinal entropy [1], we study the imbalanced case, where ordinal entropy itself cannot be used as an efficient tool to handle the imbalance. Compared to the RankSim [2,3,4], which treated all target samples equally based on the label distance, the outcome still relies heavily on the majority performance. Therefore, we have the next two stages, to enhance the median and few-shot sample performance to further improve the overall performance based on regularizing their feature Frobenius norm. Considering the fact that the majority shot samples are better learned (lower testing MAE [1,2,3,4]) than median and few-shot samples, in the second stage, we learn from the majority samples to train a linear mapping between the Frobenius norm (as dependent variable) and target labels (as independent variable). In the third stage, we use the trained linear network to predict the feature Frobenius norm of the median and few-shot samples, and we use the predictions to fine tune the previously fixed regression models. Therefore, the core contribution of this paper is to **leverage the majority sample from the pre-trained regression models to enhance the performance of the median and few-shot samples via regularizing their feature Frobenius norm.** Our experiments also further demonstrate this contribution.
>
> **W2: The ridge-based argument does not establish that enforcing global ordinal F-norms improves test MAE under imbalance**
> We admit that our analysis in the motivation part is just qualitative rather than quantitative. However, beyond our analysis in motivation, we empirically observed that the ordinal of the feature Frobenius norm on the label would be accompanied by the lower training MAE (middle part of each sub-figure in Fig.1), while the head and tail parts of each sub-figure in Fig.1 exhibit high training MAE. Therefore, we aim to regularize the feature Frobenius norm to enhance the performance on the head and tail target labels.
>
>
> **W3: The method bins the continuous label space into B bins and computes mini-batch prototypes for losses**
> Following the pioneering work in Deep Imbalanced Regression (DIR) [5], to acquire the target label $y$, the authors discretized the continuous label space into bins. Previous works [2,3,4,5] all followed this discretization of the label space and took it as a standard setup in the experiments.
>
> **W4: The three-phase schedule adds extra stages and epochs**
> The first stage of the training would cost approximate training time as the previous works [1,2,3,4,5], however, it requires fewer training epochs. The second stage of the training involves a linear layer, while the regression model part is fixed, which is computationally efficient. The last stage is to use the predicted Frobenius norm of the median and few-shot samples to fine tune the regression model. Together, the sum of the stages is equivalent to the total training epochs of the previous works [1,2,3,4,5,6], and the only additional consumption is the memory for the linear network.
>
>
> We report the average training time per mini-batch update (in seconds) for age estimation (AgeDB-DIR).
>
> |Method|AgeDB-DIR| \
> |VANILLA|13.88| \
> |LDS+FDS [5]|40.12| \
> |RankSim [2]|17.99| \
> |Ordinal Entropy [1]|18.63| \
> |SRL [6]|18.31| \
> |Ours(Stage 1)|18.05| \
> |Ours(Stage 2)|<1| \
> |Ours(Stage 3)|11.27|

---

> > ### Author Response · Authors · 2025-12-03
> > **Rebuttal 2:**
> >
> > **W5 : Reported improvements concentrate on medium/few shots, while many-shot degrades on AgeDB**
> > Enhancing the outcomes of DIR has been well-discussed in previous works, and most of them mainly rely on improving the performance of majority-shot samples. However, as [7] pointed out, improving the performance of median and few-shot samples has great medical significance. One example from [7] is a real-world healthcare task of potassium (K+) concentration regression from ECGs, where both hyperkalemia (high K+) and hypokalemia (low K+) are predominantly found in the few-shot region while normal K+ are located in the many-shot region. Hyperkalemia and hypokalemia are life-threatening conditions that can lead to cardiac arrest and ventricular fibrillation, necessitating accurate and timely detection. Conversely, normal K+ concentrations (the many-shot region) are of little concern, as inaccurate and untimely detection of these samples has minimal impact. Therefore, improving the performance of median and few-shot samples is important in real-world scenarios.
> >
> > **W6 : No statistical variability**
> > We provide the statistical variability (variance) of our experiments by using five different seeds as follows:
> > |MAE|Overall|Maj.|Med.|Few.| \
> > |AgeDB-DIR| 0.113 | 0.191 | 0.307| 0.425 | \
> > |IMDB-WIKI-DIR| 0.168 | 0.203| 0.311 | 0.444 | \
> > |STS-B-DIR| 0.098 | 0.185 | 0.293 | 0.398 | \
> >
> >
> > **W7: For STS-B-DIR, the backbone is BiLSTM+GloVe rather than modern Transformers, weakening the generality claim on NLP.**
> > For the STS-B-DIR dataset, we followed the standard settings as previous works [1,2,3,4,5,6,7] to make a fair comparison.
> >
> > **W8: confusing MSE in method**
> > We sorry for the typo, the loss we used in method is MAE instead of MSE.
> >
> > [1] Improving Deep Regression with Ordinal Entropy. ICLR 2023. \
> > [2] RankSim: Ranking Similarity Regularization for Deep Imbalanced Regression. ICML 2022 \
> > [3] Rank-N-Contrast: Learning Continuous Representations for Regression. NeurIPS 2023. \
> > [4] ConR: Contrastive Regularizer for Deep Imbalanced Regression. ICLR 2024. \
> > [5] Devolving into deep imbalanced regression. ICML 2021. \
> > [6] Improve Representation for Imbalanced Regression through Geometric Constraints. CVPR 2025. \
> > [7] Dist Loss: Enhancing Regression in Few-Shot Region through Distribution Distance Constraint. ICLR 2025.

---

### Official Review · Reviewer_JQPK · 2025-10-30

**Soundness:** 2
**Presentation:** 2
**Contribution:** 2
**Rating:** 2
**Confidence:** 4

**Summary:**

The paper addresses the problem of DIR by introducing an ordinal regularization on the Frobenius norms of features. The authors propose a three-phase training framework: training a baseline regression network + learning a mapping from labels to prototype Frobenius norms +  enforcing the model to preserve ordinality in Frobenius norms of few-shot label regions. The approach is evaluated on three benchmark datasets.

**Strengths:**

The motivation of enforcing ordinal consistency on feature scaling is interesting. The overall idea also demonstrates good empirical performance.

**Weaknesses:**

### Theoretical
- Although the linear ridge regression case suggests a correlation between label magnitude and feature scale, this relationship does not hold theoretically for deep neural networks, where the feature extractor is nonlinear and both $z$ and $\omega$ are learned jointly. It feels more like the core idea (larger label leads to larger feature norms) is just empirically observed.

- The related work [1] that the authors cite actually suggests the opposite: regression performance is sensitive to the overall feature scale, and changing it can hurt transferability. That means we usually want to keep the feature norms stable, not explicitly regularize them.

- In Phases 2 and 3, since the Frobenius norms are mainly learned from majority labels, I think the imbalance still persists.

- In standard deep networks, components like batch normalization and weight decay might destroy any monotonic relationship between label value and feature magnitude.

[1] Xinyang Chen, Sinan Wang, Jianmin Wang, and Mingsheng Long. Representation subspace distance
for domain adaptation regression. ICML 2021

### Experimental

- The method introduces multiple loss terms and training phases, so more ablations should be presented in the main paper rather than in the appendix. It’s hard to evaluate the individual contribution of each phase or loss from the current presentation.

- The improvement on STS-B-DIR is marginal. In Table 3 shows no significant improvement even on the medium-shot subset.

### Writing
The paper also contains several confusing phrases and typos, and many figures are too vague or low-resolution to read clearly.

For example:

(1) Figure 1: what metric is used here?

(2) Lines 95–96: the reference "Fig. 2 (1, 2, 3, 5, 6) in Fig. 2(4)" is unclear.

(3) "Figure 1.3", Line 124 in the appendix

**Questions:**

- Does Figure 5 in the appendix refer to the overall MAE? The proposed method mainly improves performance on medium & few-shot labels, so the analysis should focus more on these subsets.

- Ep5 + Ep6 ($\beta * L_{KL} + \beta * \alpha * L_{OE} + L_{any}$) contains hyper-parameter combinations and Fig 3 & 4 in Appendix is not enough. A more detailed 2D ablation (better in main manuscript) would better show how each term interacts.

- I think Eq. (3) is the key part of the method, but there’s no ablation that evaluates its actual effect, for example by removing it.

---

> ### Author Response · Authors · 2025-12-03
> **Rebuttal for Reviewer JQPK**
>
> **W1: It feels more like the core idea (larger label leads to larger feature norms) is just empirically observed.**
> We appreciate the reviewer’s comment. The correlation between label magnitude and feature norms is indeed empirically observed rather than theoretically derived. We treat it as a practical inductive bias supported by consistent evidence across datasets and methods. Therefore, our analysis in the Motivation section is a qualitative analysis motivated by our empirical observations.
>
> **W2 : The related work [1] usually wants to keep the feature norms stable, not explicitly regularize them.**
> Yes, we admit that the [1] is to stabilize the feature scale of the target domain as that of the source domain (aligning the two subspace base).However, our work attempts to solve another problem, which can be viewed as label shift (one aspect of domain shift), but it is not exactly the same. Firstly, in domain adaptation regression problems [1], each label is well-fitted. Compared to the label shift problem in the domain adaptation setup, deep imbalanced regression faces the problem that the median and few-shot labels are under-fitted. Therefore, [1] can view the feature norm of all labels as one collective entity. However, each label in the domain adaptation regression still exhibits \textbf{discrepancies} in their feature norms. Meanwhile, as [1] pointed out that matching distributions using instance representation has the risk of changing the feature scale, in this paper we investigate the feature norm of each label rather than the entire domain together, which does not contradict the findings of [1] and can be regarded as a more fine-grained extension of [1].
>
> **W3 : In Phases 2 and 3, since the Frobenius norms are mainly learned from majority labels, I think the imbalance still persists.**
> Our work is not to explicitly eliminate the imbalance. Instead, we try to use the well-fitted majority-shot samples to help the median and few-shot samples. Therefore, as can be observed from our experiments, the median and few-shot samples have shown significant improvements in three real-world datasets, indicating that the impacts of the imbalance on the median and few-shot samples have been significantly alleviated by our proposed method.
>
> **W4 : components like batch normalization and weight decay might destroy any monotonic relationship between label value and feature magnitude.**
> Yes, so we study the characteristic of the last-layer feature representation as we described in the Motivation section, which is not only robust enough for our regression task  [2] but also easy to implement in the experiments.
>
> **Experimental**
> We conducted the ablation study on each part of the loss in AgeDB-DIR as follows:
>
> |Loss| w/o $\mathcal{L} _ {KL}$|w/o $\mathcal{L} _ {OE}$|w/o $\mathcal{L} _ {OE}$|w/o $\mathcal{L} _ {ordinal}$|  \
> |Overall MAE (after stage 1)|7.61|7.93|7.82| \
> |Overall MAE (after stage 3)|7.52|7.71|7.68|
>
> Although the improvement of the STS-B-DIR is marginal, the reason behind this is the baseline we introduce is MAE, if we change the $\mathcal{L} _ {any}$ into any previous solutions, our method can exhibit a significant improvement under the same total training epochs.
>
>
> **Writing**
> The y-axis in Fig.1 is the MAE. We have conducted the ablation study on each part of the loss, pls refer to the Experiments.
>
> **Questions**
> Yes, it refers to the overall MAE, a more detailed improvement on the median and few-shot samples can be found in the Tables.
> We use the overall MAE here to demonstrate our empirical observation that the ordinal of feature Frobenius norm is conducive to performance.
>
>
>
> [1] Representation subspace distance for domain adaptation regression. ICML 2021. \
> [2] Last Layer Re-Training is Sufficient for Robustness to Spurious Correlations. ICLR 2023. \

---

### Official Review · Reviewer_oRTS · 2025-11-04

**Soundness:** 2
**Presentation:** 1
**Contribution:** 2
**Rating:** 2
**Confidence:** 4

**Summary:**

This paper studies the deep imbalanced regression (DIR) problem. Inspired by the influence of the Frobenius norm, the authors study the relationship of the Frobenius norm and the performance for different labels. Based on the empirical results, the authors propose a label-distance-based regularization method which focuses on the ordinality of feature Frobenius norm across the labels. The experiments on three datasets demonstrate the effectiveness of the proposed method.

**Strengths:**

- The motivation of the paper is reasonable with the empirical investigation.
- The proposed method can provide some new perspectives for the community.
- The experimental results make sense to some extent.

**Weaknesses:**

- The novelty is limited. The effect of the Frobenius norm on regression tasks is studied by the previous work. Therefore, the contribution of this paper is limited.
- The texts in the figures are too small, blurry, and quite hard to read.
- What is the meaning of y-axis of Figure 1? The authors claim that the performance of the few-shot and median-shot samples are worse than that of the majority-shot samples. However, it seems that the bars of few-shot performance are much higher than major- and medium-shots in Figure 1.
- Although the experimental results demonstrate the effectiveness on few-shot labels, the overall performance is sub-optimal. The proposed method cannot achieve best overall performance on any datasets. Unless the authors can demonstrate the importance of few-shot accuracy, which should be quite important than overall accuracy, otherwise the experimental results will be unconvincing.
- More implemental details of hyperparameters should be considered, such as loss weights, learning rate, weight decay, momentum, training epochs. It is absolutely possible that previous methods can also improve few-shot performance by modifying their hyperparameters.

[1] Representation subspace distance for domain adaptation regression.

**Questions:**

See weaknesses.

---

> ### Author Response · Authors · 2025-12-03
> **Rebuttal for Reviewer oRTS**
>
> **W1 : The novelty is limited...**
> In this paper, we leveraged the ordinal of feature Frobenius norm on the target labels to enhance the model outcome. Although previous works have studied the feature Frobenius norm [1], they have mostly studied the domain-level prototype instead of the per-label prototype. Moreover, they studied the domain adaptation cases while our study focused on the imbalanced case, where the median and few-shot samples are not well learned by the models. Therefore, we believe that our work is the first to incorporate the feature Frobenius norm in deep imbalanced regression task. More specifically, our work focused on leveraging the well-trained majority-shot samples to help the median and few-shot samples, which differs our work from other previous studies in deep imbalanced regression.
>
>
> **What is the meaning of y-axis of Figure 1?**
> Sorry for not making this point clear. The y-axis denotes the Mean-Absolute-Error(MAE), higher MAE denotes worse performance, therefore, from Fig. 1, we can observe that the higher MAE bar of the median and low-shot samples performs worse than that of the majority-shot samples.
>
> **the overall performance is sub-optimal**
> Our method aims to leverage the majority to help the median and few-shot samples. Therefore, the core of our work is to design a training framework that can effectively take advantage of the majority-shot samples via their feature Frobenius norm, resulting in a case that the improvement mostly falls in the median and few-shot samples. There is one example for explaining the importance of median few-shot accuracy from [2], real-world healthcare task of potassium (K+) concentration regression from ECGs, where both hyperkalemia (high K+) and hypokalemia (low K+) are predominantly found in the few-shot region while normal K+ are located in the many-shot region. Hyperkalemia and hypokalemia are life-threatening conditions that can lead to cardiac arrest and ventricular fibrillation, necessitating accurate and timely detection. Conversely, normal K+ concentrations (the many-shot region) are of little concern, as inaccurate and untimely detection of these samples has minimal impact. Therefore, improving the performance of median and few-shot samples is important in real-world scenarios.
>
>
> **More implemental details of hyperparameters should be considered**
> We have presented loss weights, learning rate, weight decay, momentum, and training epochs in the supplementary material and codes. Compared to previous works, our method used the average results of five seeds, which is a fair comparison to the previous works.
>
>
>
>
> [1] Representation subspace distance for domain adaptation regression. ICML 2021. \
> [2] Dist Loss: Enhancing Regression in Few-Shot Region through Distribution Distance Constraint. ICLR 2025.

---

### Official Review · Reviewer_RTFE · 2025-11-07

**Soundness:** 3
**Presentation:** 3
**Contribution:** 3
**Rating:** 6
**Confidence:** 3

**Summary:**

The paper addresses Deep Imbalanced Regression (DIR), where training data is highly skewed but testing data is balanced1. The authors observe that well-performing majority-shot samples exhibit an ordinal relationship in their feature Frobenius norms and have low training Mean-Absolute-Error (MAE), whereas underrepresented (median and few-shot) samples violate this ordinality and have high training MAE2. Motivated by this, they propose a three-phase training strategy: (1) an ordinal regularization ($\mathcal{L}_{ordinal}$) to encourage ordinal feature Frobenius norms across all labels3; (2) a majority-guided linear network ($f_F$) trained to predict these norms using only well-trained majority samples; and (3) a fine-tuning phase where the main model is regularized to match the norms predicted by $f_F$ for all samples, specifically benefiting the underrepresented ones.

**Strengths:**

Originality: The paper introduces a novel perspective on DIR by focusing specifically on the ordinality of the feature Frobenius norm as an indicator of training quality. Leveraging a simple linear predictor trained only on majority samples to guide the minority samples in a later phase is an interesting and seemingly effective strategy.

Quality: The empirical motivation is well-demonstrated in Figure 2, showing the clear divergence in norm ordinality for underperforming classes. The method is tested against a comprehensive list of standard baselines (e.g., RankSim, Con-R, SupCR) on established DIR benchmarks (AgeDB-DIR, IMDB-WIKI-DIR, STS-B-DIR).

Significance: The method achieves substantial improvements on the most challenging sub-groups. For instance, on AgeDB-DIR, it reports a 14.6% relative improvement on few-shot samples compared to the best-performing baseline SupCR (9.24 vs 10.82 MAE). It also establishes new SOTA results for few-shot and medium-shot on IMDB-WIKI-DIR.

**Weaknesses:**

Performance Trade-offs: While few-shot performance improves, there is a notable degradation in majority-shot performance in some cases. For AgeDB-DIR (Table 1), the proposed method achieves 7.35 MAE on "Many" shots, noticeably worse than SupCR (6.20) and VIR (6.39). The authors acknowledge this as a "strategic trade-off", but it raises concerns about whether strict ordinality regularization might be over-constraining the rich features of majority classes.

Procedural Complexity: The proposed method requires a three-phase training process ($e_{main} + e_F + e_{SFT}$)14141414. This is more complex to implement and tune compared to end-to-end baselines.

Theoretical Justification: The motivation in Section 3.2 relies on a simple ridge regression analysis to justify regularizing the Frobenius norm. While it serves as a decent heuristic, it may be too simplistic to fully explain the dynamics in deep non-linear networks.

**Questions:**

Could the authors elaborate further on the significant performance drop for "Many" shot samples in AgeDB-DIR (7.35 vs SupCR's 6.20)? Does forcing the majority classes to adhere strictly to the predicted ordinal norm curve prevent them from learning more complex, discriminative features they otherwise would?

Is the three-phase training strictly necessary? Specifically, could Phase 1 ($\mathcal{L}_{ordinal}$) be combined with standard training, or is the sequential nature critical for the stability of the linear predictor in Phase 2?

How sensitive is the method to the choice of hyperparameters $\alpha$ (Eq. 5) and $\beta$ (Eq. 6)?

---

> ### Author Response · Authors · 2025-12-03
> **Rebuttal for Reviewer RTFE**
>
> **W1：Performance Trade-offs ：** The goal of this work is to leverage the well-fitted majority-shot samples to help the median and few-shot samples. Therefore, during stage 2, we use the majority-shot sample feature Frobenius norm to train a linear mapping between the feature norm and teh label, and then the predicted Frobenius norm of the median and few-shot samples is used to fine tune the regression model in stage 3. Therefore, the overall performance relies heavily on the majority-shot samples.
> More importantly, the performance of majority-shot samples relies heavily on the $\mathcal{L} _ {any}$, the loss that penalizes on the regression tasks, e.g., SuperCR, RankSim, SRL, etc.
>
> Therefore, when we replace $\mathcal{L} _ {any}$ from our MAE to other methods, we have the following results:
>
> |$\mathcal{L} _ {any}$|Overall|Maj.|Med.|Few.| \
> |Ours + RankSim [1]| 6.84 | 6.32 | 7.66 | 9.35 | \
> |Ours + SuperCR [2]| |6.79 | 6.29 | 7.58 | 9.32 | \
> |Ours + OE [3]| 7.46 | 6.79 | 8.11 | 11.99 | \
> |Ours + ConR [4]| 6.89 | 6.36 | 7.76 | 9.49 |
>
> The results above show that our method can effectively and consistently improve the performance on median and few-shot samples while not significantly sacrifice the performance of majority-shot samples.
>
> **W2  : Procedural Complexity**The overall training epoch of our method is the same as the previous works [1,2,3,4], e.g., 90 epochs. The only additional bundle of our method is to train a linear network, which is light-weighted and computational efficient. Although our work introduced three sequential procedural, the overall pipeline is clear : 1) training regression model 2) training linear model 3) fine tune the regression model. Therefore, our work is easy to be implemented and does not require complex hyper-parameter tuning.
>
>
> **W3 :Theoretical Justification** We admit our work is an empirical-observation driven method. However, the empirical observation is consistent over different methods and diverse datasets, which encourages us to leverage this empirical observation to address the deep imbalanced regression. Therefore, our analysis is just qualitative rather than quantitative.
>
>
> **Q1: elaborate further on the significant performance drop for "Many" shot sample :**
> The significant drop on the "Many" is because the basic loss function is MAE, which should be compared with the baseline.
> When we change the basic loss function $\mathcal{L} _ {any}$, we can observe from the previous table that the "Many" shot has been improved significantly, which consequently improved the performance of median and few-shot samples. As a result, this also indicates the effectiveness of our proposed method.
>
> **Q2: could Phase 1 be combined with standard training?**
> Yes, the three phase is necessary.  As the quality of linear predictor relies on the performance of the "Many" shot samples, the phase 1 is to build a solid foundation for training an accurate linear network in phase 2.
>
> **Q3:** The sensitivity of the $\alpha$, $\beta$ can be found in the supplementary material.
>
>
> [1] RankSim: Ranking Similarity Regularization for Deep Imbalanced Regression. ICML 2022. \
> [2] Rank-N-Contrast: Learning Continuous Representations for Regression. NeurIPS 2023. \
> [3] Improving Deep Regression with Ordinal Entropy. ICLR 2023. \
> [4] ConR: Contrastive Regularizer for Deep Imbalanced Regression. ICLR 2024.

---

### Official Review · Reviewer_T3PB · 2025-11-08

**Soundness:** 2
**Presentation:** 2
**Contribution:** 2
**Rating:** 4
**Confidence:** 4

**Summary:**

This paper finds feature Frobenius norm and training performance have an obvious relationship, and thus, proposes a three-step training pipeline to further enhance the performance on minority-shot samples by keeping their Frobenius norm.

**Strengths:**

S1. An interesting phenomenon is obversed: the correlation between feature Frobenius norm and training performance.

S2. An effective three-step strategy to keep the ordinality of feature Frobenius norm for minority-shot samples.

S3. Experiments exhibit that the proposed method could achieve a siginitant improvement on med. and few-shot labels.

**Weaknesses:**

Major Weaknesses:

W1: In abs., the authors claim they analysize why the ordinality of the Frobenius norm can result in good training performance, but this reviewer cannot find the analysis.

W2: The proposed pipeline involves three traininig stages. It seems a little complex.

W3: The experimental results indicate that this method sacrifices the performance of majority-shot samples.

Minor Weaknesses:

W4: Unreasonable paragraph allocation. E.g., the line 75~76 is an one-sentence paragraph.

W5: low resolution and tiny fontsize of figures.

**Questions:**

Q1: why could the Frobenius norm indicate the performance?

Q2: why does the performance on major-label samples reduces?

---

> ### Author Response · Authors · 2025-12-03
> **Rebuttal for Reviewer T3PB**
>
> We sincerely appreciate your thoughtful review and insightful suggestions!
>
> **W1: In abs., the authors claim they analysize why the ordinality of the Frobenius norm can result in good training performance, but this reviewer cannot find the analysis.**
> We conducted a qualitative rather than qualitative analysis in Section 3.2 Motivation, where we formalized the correlation between the Frobenius norm and the weight vector, highlighting that maintaining the ordinal relationship of the Frobenius norm of the features would contribute to the learning of the DIR task. Furthermore, the empirical observation is consistent over different methods and diverse datasets, which encourages us to leverage this empirical observation to address the deep imbalanced regression.
>
> **W2: The proposed pipeline involves three training stages. It seems a little complex.**
> The first stage is the pre-training stage that is **algorithm-agonistic**, where the aim of the first stage is to regularize on the feature Frobenius-norm with $\mathcal{L} _ {ordinal}$ with arbitrary loss penalization $\mathcal{L} _ {any}$ on the target label $y$. The second stage is to train a tiny linear layer to learn the ordinal of the Frobenius norm only from the majority shot samples.Therefore, at this stage, the parameter of the regression model is fixed,  which is computationally efficient compared to the first stage. The third stage is to use the predicted Frobenius norm of the median and few-shot samples to fine tune the re-trained regression model. In summary, in our experiment, the total training epochs of the regression model (three stages together) are the same as in previous works, but are more computationally efficient compared to the previous works.
>
>
> **W3: The experimental results indicate that this method sacrifices the performance of majority-shot samples.**
> We acknowledge that under the loss of MAE, $\mathcal{L} _ {any} = MAE$, our method exhibits a considerably lower majority performance. However, the main contribution of this paper is to enhance the regression model using the median and few-shot samples, where improving model accuracy in the few-shot region under imbalanced data distributions continues to be both a significant challenge and an important goal in the previous works [1,2]. Meanwhile, when we use different baselines $\mathcal{L} _ {any}$ such as RankSim [3], SuperCR [4] etc, our performance significantly improves for all shots, demonstrating the effectiveness of our proposed method.
>
> **W4: Unreasonable paragraph allocation. E.g., the line 75~76 is an one-sentence paragraph.**
> Sorry for this point, we have integrated this line into the next paragraph.
>
>
> **W5: low resolution and tiny fontsize of figures.**
> We have updated the font and improved the resolution of the figures for better interpretation. The reason for this is the latex "\includegrahic" would cause blurry in rescaling, we replaced the orignal .png file with the .pdf file.
>
>
> **Q1 :  why could the Frobenius norm indicate the performance?**
> The empirical observation from the Fig.2 has demonstrated that the ordinal of fetature Frobenius norm correponds to a lower MAE, indicating the correlation between them. Our experiments further explained that controling the ordinal of feature Frobenius norm is conductive to the model performance.
>
> **Q2 : why does the performance on major-label samples reduces?**
> The performance of the majority is not reduced. The reason of the major-label reduction is the becasue the $\mathcal{L} _ {any}$ is purely MAE, when we replace the MAE with other methods, we can observe a clear improvement on the "major-label".
>
> |$\mathcal{L} _ {any}$|Overall|Maj.|Med.|Few.| \
> |Ours + RankSim [3]| 6.84 | 6.32 | 7.66 | 9.35 | \
> |Ours + SuperCR [4]| |6.79 | 6.29 | 7.58 | 9.32 | \
> |Ours + OE [5]| 7.46 | 6.79 | 8.11 | 11.99 | \
> |Ours + ConR [6]| 6.89 | 6.36 | 7.76 | 9.49 |
>
>
>
> [1] Dist Loss: Enhancing Regression in Few-Shot Region through Distribution Distance Constraint. ICLR 2025. \
> [2] Density-based weighting for imbalanced regression. Machine learning 2021. \
> [3] RankSim: Ranking Similarity Regularization for Deep Imbalanced Regression. ICML 2022. \
> [4] Rank-N-Contrast: Learning Continuous Representations for Regression. NeurIPS 2023. \
> [5] Improving Deep Regression with Ordinal Entropy. ICLR 2023. \
> [6] ConR: Contrastive Regularizer for Deep Imbalanced Regression. ICLR 2024.

---

### Meta-Review · Area_Chair_CjJM · 2025-12-27

**Summary:**

The paper addresses Deep Imbalanced Regression (DIR), where training data is highly skewed but testing data is balanced. Specifically, this paper identifies  the correlation between feature Frobenius norm and model training performance. Based on this observation, the authors propose a three-step training pipeline designed to enhance the performance on minority-shot samples by explicitly preserving their Frobenius norm.

The reviewers pointed out that while the authors claim to have analyzed how the ordinality of the Frobenius norm leads to good training performance, the explanation provided lacks sufficient persuasiveness. Most reviewers also found the paper's novelty to be limited, as the effect of the Frobenius norm in regression tasks has been studied in prior work. Furthermore, the motivation relies solely on a simple ridge regression analysis to justify norm regularization，i.e., a heuristic that may be overly simplistic for fully explaining the dynamics in deep non-linear networks. The paper also suffers from presentation issues, such as low-resolution figures and excessively small font sizes. Overall, the work lacks sufficient theoretical depth and novelty.

Considering all reviews and scores, I believe this paper in its current form does not meet the standard for publication at ICLR.

**Reviewer Concerns:**

The following reviewer conserns are outstandig:
1) The authors claim they analysize why the ordinality of the Frobenius norm can result in good training performance, but this reviewer cannot find the analysis.
2) The novelty is limited. The effect of the Frobenius norm on regression tasks is studied by the previous work.
3)  Low resolution and tiny fontsize of figures.
4) The motivation relies on a simple ridge regression analysis to justify regularizing the Frobenius norm. While it serves as a decent heuristic, it may be too simplistic to fully explain the dynamics in deep non-linear networks.

**Reviewer Scores:**

The reviewers have not responded to the authors' rebuttal; therefore, they are unlikely to revise their scores.

---

### Decision · Program_Chairs · 2026-01-26

Reject